# Prompt-based Node Feature Extractor for Few-shot Learning on Text-Attributed Graphs

## Abstract

Text-attributed Graphs (TAGs) are commonly found in the real world, such as social networks and citation networks, and consist of nodes represented by textual descriptions. Currently, mainstream machine learning methods on TAGs involve a two-stage modeling approach: (1) unsupervised node feature extraction with pre-trained language models (PLMs); and (2) supervised learning using Graph Neural Networks (GNNs). However, we observe that these representations, which have undergone large-scale pre-training, do not significantly improve performance with a limited amount of training samples. The main issue is that existing methods have not effectively integrated information from the graph and downstream tasks simultaneously. In this paper, we propose a novel framework called G-Prompt, which combines a graph adapter and task-specific prompts to extract node features. First, G-Prompt introduces a learnable GNN layer (*i.e.,* adaptor) at the end of PLMs, which is fine-tuned to better capture the masked tokens considering graph neighborhood information. After the adapter is trained, G-Prompt incorporates task-specific prompts to obtain *interpretable* node representations for the downstream task. Our experiment results demonstrate that our proposed method outperforms current state-of-the-art (SOTA) methods on few-shot node classification. More importantly, in zero-shot settings, the G-Prompt embeddings can not only provide better task interpretability than vanilla PLMs but also achieve comparable performance with fully-supervised baselines.

## 1 Introduction

Text-Attributed Graphs (TAGs) are a type of graph that have textual data as node attributes. These types of graphs are prevalent in the real world, such as in citation networks [12] where the node attribute is the paper's abstract. TAGs have diverse potential applications, including paper classification [3] and user profiling[14]. However, studying TAGs presents a significant challenge: how to model the intricate interplay between graph structures and textual features. This issue has been extensively explored in several fields, including natural language processing, information extraction, and graph representation learning.

An idealized approach involves combining pre-trained language models (PLMs) [10, 20] with graph neural networks and jointly training them [35, 24]. Nevertheless, this method requires fine-tuning the PLMs, which demands substantial computational resources. Additionally, trained models are hard to be reused in other tasks because finetuning PLM may bring catastrophic forgetting[2].

Therefore, a more commonly used and efficient approach is the two-stage process [32, 34, 23]: (1) utilizing pre-trained language models (PLMs) for unsupervised modeling of the nodes' textual features. (2) supervised learning using Graph Neural Networks (GNNs). Compared to joint training of PLMs and GNNs, this approach offers several advantages in practical applications. For example,

it can be combined with numerous GNN frameworks or PLMs, and this approach does not require fine-tuning PLMs for every downstream task. However, PLMs are unable to fully leverage the wealth of information contained in the graph structure, which represents significant information. To overcome these limitations, some works propose self-supervised fine-tuning PLMs using graph information to extract graph-aware node features [3]. Such methods have achieved significant success across various benchmark datasets[12].

However, both self-supervised methods and using language models directly to process TAG suffer from a fundamental drawback. They cannot incorporate downstream task information, which results in identical representations being generated for all downstream tasks. This is evidently counterintuitive as the required information may vary for different tasks. For example, height is useful information in predicting a user's weight but fails to accurately predict age. This issue can be resolved by utilizing task-specific prompts combined with language models [26] to extract downstream task-related representations. For example, suppose we have a paper's abstract $\{\mathbf{Abstract}\}$ in a citation network, and the task is to classify the subject of the paper. We can add some prompts to a node's sentence: $\{This, is, a, paper, of, [\mathbf{mask}], subject, its, abstract, is, :, \mathbf{Abstract}\}$. And then use the embedding corresponding to the [mask] token generated by PLMs as the node feature. Yet this approach fails to effectively integrate graph information.

To better integrate task-specific information and knowledge within graph structure, this paper proposes a novel framework called G-Prompt. G-Prompt combines a graph adapter and task-specific prompts to extract node features. Specifically, G-Prompt contains a graph adapter that helps PLMs become aware of graph structures. This graph adapter is self-supervised and trained by fill-mask tasks on specific TAGs. G-Prompt then incorporates task-specific prompts to obtain interpretable node representations for downstream tasks.

We conduct extensive experiments on three real-world datasets in the domains of few-shot and zero-shot learning, in order to demonstrate the effectiveness of our proposed method. The results of our experiments show that G-Prompt achieves state-of-the-art performance in few-shot learning, with an average improvement of *avg.* 4.1% compared to the best baseline. Besides, our G-Prompt embeddings are also highly robust in zero-shot settings, outperforming PLMs by *avg.* 2.7%. Furthermore, we conduct an analysis of the representations generated by G-Prompt and found that they have high interpretability with respect to task performance.

## 2 Background

### 2.1 Text-Attributed Graph

Let $G = \{V, A\}$ denotes a text-attributed graph (TAG), where $V$ is the node set and $A$ is the adjacency matrix. Each node $i \in V$ is associated with a sentence $S_i = \{s_{i,0}, s_{i,1}, ..., s_{i,|S_i|}\}$, which represents the textual feature of the node. In most cases, the first token in each sentence (i.e., $s_{i,0}$) is [**cls**], indicating the beginning of the sentence. This paper focuses on how to unsupervised extract high-quality node features on TAGs for various downstream tasks.

### 2.2 Pretrained Language Models

Before we introduce G-Prompt, we require some basic concepts of pre-trained language models.

**Framework of PLMs**. A PLM consists of a multi-layer transformer encoder that takes a sentence $S_i$ as input and outputs the hidden states of each token:

$$\mathbf{PLM}(\{s_{i,0}, s_{i,1}, ..., s_{i,|S_i|}\}) = \{h_{i,0}, h_{i,1}, ..., h_{i,|S_i|}\}, \tag{1}$$

where $h_{i,k}$ is the dense hidden state of $s_{i,k}$.

**Pretraining of PLMs**. The fill-mask task is commonly used to pre-train PLMs [4, 20, 10]. Given a sentence $S_i$, the mask stage involves randomly selecting some tokens and replacing them with either [**mask**] or random tokens, resulting in a modified sentence $\hat{S}_i = \{s_{i,0}, s_{i,1}, ..., \hat{s}_{i,k}, ..., s_{i,|S_i|}\}$, where $\hat{s}_{i,k}$ represents the masked token. In the filling stage, $\hat{S}_i$ is passed through the transformer encoder, which outputs the hidden states of each token. We denote the hidden state of the masked token $\hat{s}_{i,k}$ as $\hat{h}_{i,k}$, which is used to predict the ID of the masked token:

$$\hat{y}_{i,k} = f_{\mathrm{LM}}(\hat{h}_{i,k}), \tag{2}$$

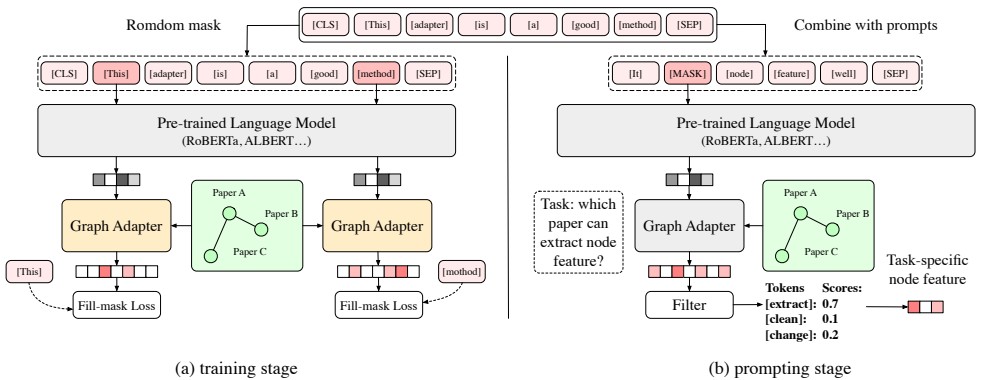

Figure 1: Framework of G-Prompt

where $f_{LM}$ is a linear transformation with softmax fuction, $\hat{y}_{i,k} \in \mathbb{N}^{1 \times T}$, and $T$ is the size of the vocabulary. The loss function of the fill-mask task is defined as $\mathcal{L} = \mathrm{CE}(\hat{y}_{i,k}, y_{i,k})$, where $y_{i,k}$ is the ID of the masked token, and $\mathrm{CE}(\cdot, \cdot)$ is the cross-entropy loss.

**Sentence Embedding**. The hidden state of the [**cls**] token ($h_{i,0}$) and the mean-pooling of all hidden states are commonly used as sentence embeddings [28, 6].

**Prompting on PLMs**. Sentence embedding and token embedding are simultaneously pre-trained in many PLMs. However, due to the gap between pretraining tasks and downstream tasks, sentence embedding always requires fine-tuning for specific tasks. To address this issue, some studies utilize prompts to extract sentence features [13]. For example, suppose we have a paper's abstract {**Abstract**}, and the task is to classify the subject of it. We can add some prompts to the sentence:

$$\{This, is, a, paper, of, [\mathbf{mask}], subject, its, abstract, is, :, \mathbf{Abstract}\} \tag{3}$$

Then this sentence is encoded by PLMs, and we let $h_{i|p}$ denote the hidden state of the [**mask**] token in prompts. Extensive experiment shows that using prompts can shorten the gap between PLMs and downstream tasks and maximize the utilization of the knowledge PLMs learned during pretraining.

## 2.3 Graph Neural Networks

Graph Neural Networks (GNNs) have achieved remarkable success in modeling graph-structured data[30, 7]. The message-passing framework is a commonly used architecture of GNN. At a high level, GNNs take a set of node features $X^0$ and an adjacency matrix $A$ as input and iteratively capture neighbors' information via message-passing. More specifically, for a given node $i \in V$, each layer of message-passing can be expressed as:

$$x_i^k = \mathbf{Pool}\{f_\theta(x_j^{k-1})|j \in \mathcal{N}_i\} \tag{4}$$

where $\mathbf{Pool}\{\cdot\}$ is an aggregation function that combines the features of neighboring nodes, such as mean-pooling. And $\mathcal{N}_i$ denotes the set of neighbors of node $i$.

## 3 Method: G-Prompt

Utilizing the information of downstream tasks and graphs is crucial for generating high-quality node representations. The term "high quality" is inherently task-specific, as exemplified by the fact that height is a useful feature in predicting user weight but fails to accurately predict age. Besides, the valuable topological information of TAGs can significantly enhance the understanding of textual features in TAGs. However, extracting node features using both task and graph information simultaneously is significantly challenging. Current PLMs used for handling textual features are graph-free, while current graph-based methods employed to extract node features are primarily task-free. Therefore, this paper proposes a novel self-supervised method, G-Prompt, capable of extracting task-specific and graph-aware node representations.

## 3.1 Overview

While previous works have frequently employed PLMs to process TAGs, these investigations have been constrained in extracting a broad node representation from the text-based characteristics and have not incorporated task-specific prior knowledge. Consequently, additional learning supervision via GNNs is needed to enable the effective adaptation of these node representations to downstream tasks. To address this limitation, the paper suggests incorporating prompts and PLMs into the process of extracting task-relevant node features from TAGs. Nevertheless, PLMs only utilize contextual information to generate the prompts-related output, which may be insufficient for handling TAGs. Graph structures often contain essential information that can facilitate a better understanding of textual features. For instance, in a citation network, a masked sentence such as *"This paper focuses on [MASK] learning in AI domain"* could have multiple candidate tokens based solely on context. However, if many papers related to graphs are cited, we can infer with greater confidence that the masked token is likely *"graph"*. At present, PLMs operate solely based on context, and their structure is graph-free. Directly incorporating graph information into PLMs by prompts is not feasible because limited prompts cannot describe the entire topological structure adequately.

Therefore, the proposed G-Prompt leverages a self-supervised based graph adapter and prompts to make PLMs aware of the graph information and downstream task. Given a specific TAG, the pipeline of G-Prompt is as follows: (1) Training an adapter on the given TAG to make PLMs graph-aware. Specifically, we propose a graph adapter that operates on the prediction layer of PLMs to assist in capturing graph information, which is fine-tuned by the fill-mask task based on the textual data contained by the given TAG. (2) Employing task-specific prompts and fine-tuned graph adapters to generate task-aware and graph-aware node features.

## 3.2 Fine-Tuning PLMs with the Graph Adapter

Using adapters to enable PLMs to perceive graph information is a straightforward idea. However, unlike adapters used for downstream task fine-tuning [11, 18], the graph adapter is used to combine prompts in order to extract task-relevant node representations. This is an unsupervised process, which means that the graph adapter only receives self-supervised training on given TAGs. Consequently, the most challenging aspect of graph adapters is how to assist PLMs in perceiving graph information while also maintaining their contextual understanding capability. Additionally, the graph adapter is only trained on a given TAG, generalizing to prompt tokens can also be quite difficult. Next, we introduce the graph adapter and discuss how it overcomes these challenges in detail.

**Context-friendly adapter placement.** The fill-mask task involves two modules of PLMs: a transformer-based module that models context information to obtain representations of masked tokens and a linear transformation that decodes the representation to output the probable IDs of the masked token. To avoid compromising the contextual modeling ability of PLMs, the Graph Adapter only perform on the last layer of PLMs. More specifically, the graph adapter is a GNN structure combing with the pre-trained final layer of the PLMs. Given a specific masked token $\hat{s}_{i,k}$, The inputs of the Graph Adapter are the masked token $\hat{h}_{i,k}$, sentence representations of node $i$ and its neighbors and output is the prediction of the IDs' of the masked token. This process aligns with intuition — inferring a possible token based on context first and then determining the final token based on graph information. Formally,

$$\hat{y}_{i,k} = \textbf{GraphAdapter}\{f_{\text{LM}}, \hat{h}_{i,k}, z_i, \{z_j \in \mathcal{N}_i\}, \Theta\}, \tag{5}$$

where the $z_i$ and $z_j$ denote the sentence embedding of node $i$ and $j$. Note, sentence embedding is task-free and only used to model nodes' influence on their neighbor. In this paper, we utilize sentence embedding of nodes' textual features as their node feature. $\Theta$ is all trainable parameters of the Graph Adapter.

**Prompting-friendly network structure**. The parameters of the adapter are only trained on the fill-mask task based on the textual data contained by the target TAG. But the adapter will be used for combining prompts to generate task-related node features in various subsequent downstream tasks. So the generalization ability of the adapter is crucial. On the one hand, the distribution of hidden states responding to masked tokens in prompts may be different from the hidden states used to train the adapter. On the other hand, the candidate tokens for task-specific prompts may not appear in the tokens of the TAG. Therefore, we carefully design the network structure of the graph adapter and utilize the pre-trained prediction layer of PLM to improve the generalization ability of it.

When it comes to the graph adapter's training stage, it's possible that the hidden states associated with certain prompts may not be present. This means that directly manipulating those hidden states could result in overfitting on the tokens already present in the TAGs. Therefore, the graph adapter models the influence of each modeled node on the distribution of surrounding neighbor tokens based on node feature, which remains unchanged when prompts are added. Considering that some tokens can be predicted well based solely on their context and that different neighbors may have different influences on the same node, the impact of a neighbor on a token is determined jointly by a gate mechanism and the token's context. Give a specific node $i$, it's neighbor $j$, and hidden states of a masked token $\hat{h}_{i,j}$,

$$\tilde{h}_{i,k,j} = a_{ij}\hat{h}_{i,k} + (1 - a_{ij})g(z_j, \Theta_g) \tag{6}$$

where $a_{ij} = \text{sigmoid}((z_i W_q)(z_j W_k)^T)$. Here, $g(\cdot)$ represents multi-layer perceptions (MLPs) with parameters $\Theta_g$ that model the influence of node $j$. It is worth noting that when considering the entire graph, $g(z_j, \Theta_g)$ will be combined with many marked tokens of node $j$'s neighbors, which can help to prevent $g(z_j, \Theta_g)$ from being overfitted on a few tokens.

Subsequently, the graph adapter combines all neighbor influence to infer the final prediction result. Since the prediction layer of PLM (i.e., $f_{LM}(\cdot)$) is well-trained on massive tokens, it also contains an amount of knowledge. Therefore, the graph adapter reuses this layer to predict the final result.

$$\tilde{y}_{i,k} = \textbf{Pool}\{f_{\text{LM}}(\tilde{h}_{i,k,j})|j \in \mathcal{N}_i\}, \tag{7}$$

In this equation, the $\textbf{Pool}(\cdot)$ used in this paper is mean-pooling. It is worth noting that the position of $f_{\text{LM}}(\cdot)$ can be interchanged with pooling since it is just a linear transformation. All trainable parameters in the graph adapter are denoted by $\Theta = \{\Theta_g, W_q, W_k\}$.

## 3.3 Model optimization of G-Prompt

The graph adapter is optimized by the original fill-mask loss, $\mathcal{L}_{i,k} = \text{CE}(\tilde{y}_{i,k}, y_{i,k})$, where $\hat{y}_{i,k}$ is the predicted probability of the $k$-th masked token for the node $i$ and $y_{i,k}$ is the true label. We aim to minimize $\mathcal{L}_{i,k}$ with respect to $\Theta$.

However, in actual optimization, the prediction results of $\tilde{y}_{i,k,j} = f_{\text{LM}}(\tilde{h}_{i,k,j})$ consist of many small values because of the large vocabulary size of the language model. Therefore, using mean-pooling presents a significant problem as it is insensitive to these small values. For example, during some stages of the optimization process, a node may have mostly $0.9$ predictions for the ground truth based on each edge, with only a few being $0.1$. Averaging them together would result in a very smooth loss, making it difficult to train the influence of neighbors with temporarily predicted values of $0.1$. To address this issue, we use geometric mean instead of mean-pooling in the finetuning stage of the graph adapter, which is more sensitive to small values. It is easy to prove that the geometric mean of positive numbers is smaller than the arithmetic means, making it harder to smooth and helping the model converge faster. formally, in finetuning stage, the loss function is:

$$\mathcal{L}_{i,k} = -y_{i,k} \odot \log\{(\prod_{j \in \mathcal{N}_i} \tilde{y}_{i,k,j})^{1/|\mathcal{N}_i|}\} = -\sum_{j \in \mathcal{N}_i} \frac{1}{|\mathcal{N}_i|} y_{i,k} \odot \log(\tilde{y}_{i,k,j}) \tag{8}$$

On the right-hand side of the equation, we are essentially minimizing $\tilde{y}_{i,k,j}$ through the cross-entropy loss $\mathcal{L}_{i,k,j} = \frac{1}{|\mathcal{N}_i|}\text{CE}(\tilde{y}_{i,k,j}, y_{i,k})$. It is worth noting that the graph adapter is only performed on the last layer of PLMs. As a result, we can sample a set of masked tokens and preserve their hidden states inferred by the PLMs before training. This implies that training of graph adapters can be achieved with very few computing resources.

## 3.4 Prompt-based Node Representations

After training the graph adapter, it can be combined with task-specific prompts to generate task-specific and graph-aware node representations. Similar to other prompt-based approaches, we simply add task-specific prompts directly into the textual feature. For example, we might use the prompt "This is a [MASK] user, consider their profile: [textual feature]." Formally, this process can be expressed as $\hat{h}_{i|p} = \textbf{PLM}(\{[P_0], [P_1]...[MASK], S_i\})$. Where, $\hat{h}_{i|p}$ represents the hidden state of the inserted [MASK], while $[P_0], [P_1]...$ refers to the task-specific prompts. The resulting hidden state is then fed into the graph encoder to decode the most probable token.

$$\hat{y}_{i|p} = \textbf{Pool}\{f_{\text{LM}}(a_{i,j}\hat{h}_{i|p} + (1 - a_{i,j})g(z_j, \Theta_g))|j \in \mathcal{N}_i\} \tag{9}$$

Table 1: Statistics of the datasets

| Dataset | # Nodes | # Eeges | Avg. Node Degree | Test Ratio (%) | Metric |
|---|---|---|---|---|---|
| **Arxiv** | 169,343 | 1,166,243 | 13.7 | 28 | ACC |
| **Instagram** | 11,339 | 377,812 | 66.6 | 60 | ROC |
| **Reddit** | 33,434 | 198,448 | 11.9 | 33 | ROC |

$\hat{y}_{i|p}$ is a $|D|$-dimensional vector, where $|D|$ is the size of the PLM vocabulary. Therefore, directly using this prediction result for node representation is not conducive to downstream tasks and storage. Thus, we use the filtered results as node features, denoted by $x_{i|p} = \text{Filter}(\hat{y}_{i|p})$. Note, each dimension represents the probability of a token being inferred by PLMs with the graph adapter based on node textual features, neighbors' information, and task-respected prompts. Intuitively, tokens that are unrelated to downstream tasks are almost the same for all nodes. Therefore, for $Y_p \in \mathbb{N}^{|V| \times |D|}$, which denotes prediction results of all nodes. This paper sorts all columns of $Y_p$ in descending order of standard deviation and keeps the top $M$ columns as the node features. Note, we can also manually select task-relevant tokens based on prior knowledge of the task and use them as node features.

# 4 Experiment

## 4.1 Experiment setup

**Dataset.** We conduct experiments on three public and real-world datasets, which are Ogbn-arxiv[12] (shorted as Arxiv), Instagram[14], and Reddit[1], to evaluate the effectiveness of the proposed method G-Prompt. Specifically, Ogbn-arxiv is a citation network where edges represent citation relationships, nodes represent papers and the text attribute is the abstracts of papers. The task is to predict paper subjects. Instagram is a social network where edges represent following relationships, nodes represent users, and the prediction task is to classify commercial users and normal users in this network. The text attribute is the users' profile. Reddit is also a social network where each node denotes a user, the node features are the content of users' historically published subreddits, and edges denote whether two users have replied to each other. The prediction task is to classify whether a user is in the top 50% popular (average score of all subreddits). Table 1 shows detailed statistics of these datasets. More details about Instagram and Reddit are provided in the Appendix.

**Compared methods.** We compare the proposed G-Prompt with PLM-based and Graph-based node feature-extracting methods. For the PLM-based methods, we consider three options: (1) direct use of sentence embedding as node features, and (2) use of the hidden states of masked tokens based on hard prompts as node features. (3) use of the prediction result of masked tokens based on prompts as node feature. For graph-based methods, we compare our proposed method with GAE and GIANT, which first conduct self-supervised learning on graphs to train PLMs or node feature encoders. To ensure a fair comparison, we add prompts into graph-based baselines. Except for GAINT and OGB features, the PLM we use in this paper is RoBERTa-Large[20]. Note that all prompts used in baselines are the same as those in G-Prompt.

**Implementation details.** For G-Prompt, we first train three graph adapters of G-Prompt on Arxiv, Instagram, and Reddit with 50 epochs, 100 epochs, and 100 epochs respectively. All of them are optimized using AdamW[21] with warm-up. For more details on the hyper-parameter settings, please refer to the Appendix. For each node, we replace 10% tokens with [mask] and use these masked tokens to train the graph adapter. During the whole training stage, all task-related prompts are invisible. Then we use prompts, finetuned graph adapters, and PLMs to jointly extract node features. For graph-based methods, we train them on each dataset with searched hyper-parameters.

## 4.2 Few-shot learning

To evaluate the performance of representations generated by different methods in few-shot learning, we compare the performance of different representations at different shot numbers based on the same GNN backbone. The GNN backbone used in the performance comparison on different shot numbers is GraphSAGE[30]. In addition, we also compare the performance of different representations combined with three different neural network architectures (i.e., MLP, and RevGAT[17]) on downstream tasks

---

[1]`https://convokit.cornell.edu/documentation/subreddit.html`

Table 2: The performance in different shots on three datasets

| Dataset # shots per class | Arxiv | | | Instagram | | | Reddit | | |
|---|---|---|---|---|---|---|---|---|---|
| | 10 | 50 | 100 | 10 | 50 | 100 | 10 | 50 | 100 |
| OGB-Feature | 0.4576 ±0.0324 | 0.5495 ±0.0171 | 0.5875 ±0.0146 | - | - | - | - | - | - |
| PLM+GAE | 0.5016 ±0.0510 | 0.5608 ±0.0101 | 0.5810 ±0.0125 | 0.5258 ±0.0635 | 0.5818 ±0.0101 | 0.5821 ±0.0058 | 0.5653 ±0.0256 | 0.6019 ±0.0174 | 0.6154 ±0.0128 |
| PLM+GAE+prompt | 0.5189 ±0.0333 | 0.5801 ±0.0102 | 0.6063 ±0.0109 | 0.5418 ±0.0298 | 0.5705 ±0.0233 | 0.5867 ±0.0100 | 0.5619 ±0.0425 | 0.5968 ±0.0237 | 0.6173 ±0.0160 |
| GIANT | 0.5050 ±0.0308 | 0.5798 ±0.0119 | 0.6081 ±0.0109 | 0.5185 ±0.0323 | 0.5601 ±0.0304 | 0.5752 ±0.0251 | 0.5618 ±0.0431 | 0.5954 ±0.0131 | 0.6130 ±0.0117 |
| GIANT + prompt | 0.5140 ±0.0320 | 0.5809 ±0.0223 | 0.6126 ±0.0159 | 0.5239 ±0.0309 | 0.5721 ±0.0361 | 0.5949 ±0.0089 | 0.5661 ±0.0459 | 0.5968 ±0.0096 | 0.6145 ±0.0105 |
| PLM-cls | 0.4697 ±0.0577 | 0.5414 ±0.0400 | 0.5869 ±0.0300 | 0.5165 ±0.0217 | 0.5385 ±0.0344 | 0.5690 ±0.0253 | 0.4965 ±0.0373 | 0.5236 ±0.0394 | 0.5754 ±0.0348 |
| PLM-Prompt-dense | 0.5117 ±0.0398 | 0.5631 ±0.0352 | 0.5865 ±0.0296 | 0.5458 ±0.0420 | 0.5796 ±0.0276 | 0.6055 ±0.0122 | 0.5363 ±0.0530 | 0.5648 ±0.0385 | 0.5998 ±0.0383 |
| PLM-Prompt-sparse | 0.5201 ±0.0284 | 0.5784 ±0.0213 | 0.6085 ±0.0203 | 0.5363 ±0.0348 | 0.5757 ±0.0225 | 0.5910 ±0.0229 | 0.5403 ±0.0424 | 0.5761 ±0.0359 | 0.6082 ±0.0192 |
| G-Prompt | 0.5248±0.0382 | **0.5927 ±0.0142** | 0.6167 ±0.0138 | **0.5576 ±0.0330** | **0.5917 ±0.0242** | **0.6090 ±0.0135** | **0.5728 ±0.0491** | **0.6167 ±0.0289** | **0.6472 ±0.0224** |
| G-Prompt w/o gate | **0.5291 ±0.0315** | 0.5877 ±0.0192 | **0.6212 ±0.0190** | 0.5507±0.0336 | 0.5706 ±0.0262 | 0.5942 ±0.0178 | 0.5501 ±0.0604 | 0.5926±0.0385 | 0.6361±0.0268 |
| G-Prompt w/o graph | 0.5226 ±0.0322 | 0.5880±0.0168 | 0.6059 ±0.0101 | 0.5234 ±0.0236 | 0.5657 ±0.0377 | 0.5914 ±0.0199 | 0.5536 ±0.0438 | 0.5683 ±0.0390 | 0.6054 ±0.0263 |
| G-Prompt w/o SSL | 0.5210 ±0.0372 | 0.5793 ±0.0168 | 0.6092 ±0.0168 | 0.5378 ±0.0419 | 0.5801±0.0269 | 0.6004±0.0193 | 0.5494 ±0.0502 | 0.5885 ±0.0365 | 0.6149 ±0.0263 |

with the same number of shots. For Arxiv, we use a publicly available partitioned test set, while for Instagram and Reddit, we randomly sample 60% and 33% of the data as the test sets, respectively. To consider the randomness of partitioning and training, each experimental result is based on five random partitions (the partitions are the same for different baselines), the experiment is repeated five times for each partition, and the variance of 5×5 results is reported.

The experiment results on different shots-num are shown in Table 2. The experiment shows that: (1) **Graph-aware can improve the performance of node representation**. In general, approaches that use sentence representations or those that involve self-supervised training with graph information tend to outperform non-trained representations. For example, GAE shows an average improvement of *avg.* 6.2% compared to RoBERTa's [cls], and GIANT shows *avg.* 6.2% improvement over cls representation. For graph-based self-supervised tasks, fine-tuning language models might be more suitable for larger datasets. GIANT outperforms GAE by *avg.* 3.0% on Arxiv, but lags behind by *avg.* 1.4% on Instagram and Reddit. (2) **Downstream task-related prompts can improve performance for all models**. For graph-free language models, prompt-based representations can improve performance by *avg.* 5.7%, and the overall performance of prediction values and hidden states corresponding to prompts is similar. For graph-based methods, prompts in GAE improve performance by *avg.* 1.3%, while prompts in GIANT lead to an average improvement of *avg.* 1.2%. However, we note that prompts are unstable for graph-based pre-trained models. GAE shows a decline in 4 experiments, while prompts only bring a slight improvement in GIANT (compared to language models). (3) **Our method is capable of utilizing both graph perception and downstream task prompts simultaneously**, achieving state-of-the-art performance. Compared to PLM representations without prompts, our method improves by *avg.* 10.6%. Compared to PLM-prompt, it improves by *avg.* 4.6%, and compared to GIANT, it improves by *avg.* 4.1%.

Besides, as Figure 2 shows, the node representation extracted by G-Prompt in different GNN-backbone also achieves the SOTA performance compared to other baseline methods.

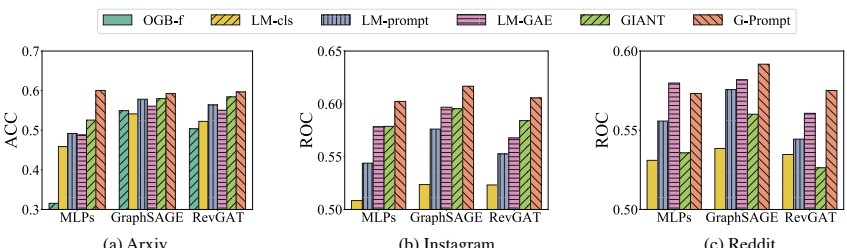

Figure 2: Comparison with different GNN backbone on 50-shots setting

### 4.3 In-depth analysis of G-Prompt

To validate the rationality of G-Prompt, we conduct experiments to compare the performance of G-Prompt and its variants. These variants include removing the gate mechanism in graph-adapter (denoted as "w/o gate"), keeping only self-loops while removing the input graph (denoted as "w/o graph"), and not training graph-adapter by self-supervised learning (denoted as "w/o SSL"). The experimental results show that all variants perform worse than G-Prompt. Specifically, removing the Graph-Adapter training process leads to *avg.* 2.8% decrease in performance, which demonstrates the effectiveness of training graph-adapter through the fill- mask task. After removing the graph input, the performance of G-Prompt decreases by *avg.* 3.8%, which further confirms that the improvement provided by G-Prompt, compared to using language model prompts directly, stems from the graph adapter's ability to assist language models in comprehending graph structures. Moreover, removing the gate mechanism results in a *avg.* 1.8% decrease in performance, indicating that the design of the graph-adapter structure is reasonable.

### 4.4 Zero-shot node classification and interpretability

The node features generated through GPrompt represent the probability of each possible word for nodes given task-related prompts, where each dimension corresponds to a specific word. This probability generation incorporates prior knowledge from PLMs, graph information, and node context. Two natural questions arise: **How much knowledge is contained within this word probability? Whether the node feature can help us interpret the downstream task?** Therefore, we further conduct zero-shot node classification experiments on node representations. Meanwhile, we conduct a case study on Instagram.

**Zero-shot node classification.** We select different sets of candidate words and sum up the probabilities of each word in the set to obtain the prediction result for node classification. We employ the ROC as the evaluation metric to assess the performance of node classification. For simplicity, ArXiv dataset only selects two categories, "Artificial Intelligence" and "Linguistics and Language". The other two datasets remain unchanged. We select completely random, bag-of-words, RoBERTa-base, and RoBERTa-large as baselines, using the same prompts as G-Prompt for PLMs. We provide experiment results of G-Prompt based on RoBERTa-base (Ours-B) and RoBERTa-large (Ours-L).

According to the results shown in Table 3. (1) The bag-of-words method has almost no predictive ability. (2)The PLM through Prompts has predictive ability on different tasks (improvement compared to BOW by *avg.* 13%). But there is a performance difference between base and large even with the same prompt due to the sensitivity of language models to prompts [22]. (3) Compared to a language model, G-Prompt shows significant performance improvement. Specifically, G-Prompt-base improved *avg.* 2.7% compared to the language model. However, it should be noted that the basic predictive ability of the language model and G-Prompt are correlated. Specifically, the correlation coefficient between the results of GPrompt-L and LM-L is 0.64, while the correlation coefficient with LM-B is 0.84. (4) Moreover, selecting more candidate words through prior knowledge can effectively help G-Prompt improve its zero-shot capability, with an average improvement of *avg.* 4.8% for the base and *avg.* 5.3% for the large. However, there is no significant improvement for language models and bag-of-words. Surprisingly, by adding a small number of candidate words, G-Prompt's zero-shot performance is already close to or even sometimes surpasses supervised training with 100 shots. This result indicates that combining language models and graphs for zero-shot learning on TAG is feasible.

**Interpretability.** The task on Instagram is to determine whether a node is a commercial user. We use the probability corresponding to each token as the prediction value, calculate its corresponding ROC of prediction performance, and then display the top 7 tokens with the highest scores. For comparison, we also show the scores of tokens corresponding to RoBERTa-Large under the same prompt. Overall, the top 7 tokens given by our model have considerably higher ROC scores than RoBERTa-Large resulting in *avg.* 7.0% improvement. Additionally, our results are intuitive and can even help explain the task, for example, "premium." Based on this result, we search and find that there are "premium creator subscriptions" on Instagram, which means "Users can set their own prices and earn extra cash each month,"[2] and this information is indeed related to commercial activity. Similarly, "niche" is also a word related to Instagram business behavior.

---

[2] https://www.pcmag.com/news/instagram-introduces-premium-creator-subscriptions

Table 3: The performance of different models in zero-shot learning

| Dataset | Pos. vocab | Neg. vocab | Rand. | BOW | LM-B | LM-L | Ours-B | Ours-L | 100 shot. |
|---|---|---|---|---|---|---|---|---|---|
| **Arxiv** | {*intellectual*} | {*language*} | 0.5021 ±0.0124 | 0.4994 ±0.0000 | 0.5955 ±0.0000 | 0.6747 ±0.0000 | 0.5840 ±0.0000 | **0.6765**$^*$ **±0.0000** | 0.9040 ±0.0253 |
| | {*intellectual, decision, logic, ...*} | {*language, translation, speech, ...*} | 0.4988 ±0.0139 | 0.5474 ±0.0000 | 0.6284 ±0.0000 | 0.6075 ±0.0000 | 0.6006 ±0.0000 | **0.7064**$^*$ **±0.0000** | |
| **Instagram** | {*commercial*} | {*normal*} | 0.5004 ±0.0151 | 0.5001 ±0.0007 | **0.5509**$^*$ **±0.0163** | 0.5365 ±0.0054 | 0.5403 ±0.0078 | 0.5382 ±0.0095 | 0.5690 ±0.0253 |
| | {*commercial, sponsored, brand, ...*} | {*normal, personality, private, ...*} | 0.5007 ±0.0131 | 0.5022 ±0.0008 | 0.5586 ±0.0117 | 0.5577 ±0.0068 | **0.5995**$^*$ **±0.0074** | 0.5957 ±0.0081 | |
| **Reddit** | {*pretty*} | {*simple*} | 0.5034 ±0.0073 | 0.5053 ±0.0019 | 0.5608 ±0.0050 | 0.5352 ±0.0027 | 0.5630 ±0.0082 | **0.5673**$^*$ **±0.0070** | 0.5754 ±0.0348 |
| | {*pretty, hilarious, funny, ...*} | {*simple, anonymous, standard, ...*} | 0.4990 ±0.0042 | 0.5034 ±0.0017 | 0.5604 ±0.0081 | 0.5587 ±0.0052 | 0.5674 ±0.0058 | **0.5742**$^*$ **±0.0066** | |

Table 4: Top 7 Tokens related to predicting commercial users on Instagram

| RoBERTa-large | | G-Prompt | |
|---|---|---|---|
| **Top 7 tokens** | **ROC** | **Top 7 tokens** | **ROC** |
| *critical* | 0.546 | *special* | 0.592 |
| *convenient* | 0.542 | *convenient* | 0.579 |
| *terrific* | 0.542 | *premium* | 0.579 |
| *banner* | 0.542 | *unique* | 0.577 |
| *gateway* | 0.539 | *great* | 0.575 |
| *compelling* | 0.539 | *pioneer* | 0.575 |
| *neat* | 0.538 | *niche* | 0.575 |

# 5 Related work

Modeling TAGs involves numerous works related to the NLP domain and Graph domain. Currently, pre-trained language models are the primary method for modeling the textual information in text-as-graphs [25]. Presently, pre-trained language models are mainly based on transformer structures[29], with a variety of pre-training methods, such as fill-mask [4], paragraph prediction[4], adversarial learning[10], and auto-regressive learning[27]. Based on these tasks, many excellent pre-trained models have emerged, including BERT[4], RoBERTa[20], and GPT3[1]. PLMs contain an amount of knowledge acquired through extensive pre-training data[31]. Recently, using prompts has been proposed to better utilize the performance of pre-trained language models[1]. Based on this finding, prompt learning[19, 8, 16] has achieved impressive results in few-shot and zero-shot learning and has been widely applied by other domains. Currently, the structural information in modeling TAGs is primarily modeled through GNNs, such as GraphSAGE[9], GAT[30], APPNP[7, 5] and RevGAT[17], and there are also many pre-training tasks on graphs such as GAE[15], GraphCL[33] that can be extended to TAGs. Recently, many methods explore better utilizing the knowledge of PLMs to model TAGs more effectively, such as pre-training language models through graph-related tasks [3] and finetuning PLMs together with GNNs via knowledge distillation[24] or variational inference [35].

# 6 Conclusion

This paper proposes G-Prompt to fuse PLMs and Graphs for extracting task-specific and graph-aware node representation in TAGs. G-Prompt have two-stage: (1) self-supervised train a graph adapter to make PLMs graph-aware based TAGs, and (2) employing prompts with the trained graph adapter to extract node representation from TAGs. Experiments with different shot settings using three datasets demonstrate that the proposed model can effectively capture both text and graph information, resulting in improved performance for few-shot learning. In zero-shot learning, our model achieves comparable performance with supervised baselines and has huge potential for future work. Furthermore, our model provides useful interpretations, which is essential for understanding the tasks and TAGs.

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
