# A  Appendix

## A.1  Dataset details

**Arxiv**. This paper uses the public partition, ground truth, and text information provided by OGB[12]. The few-shot train samples are sampled from the train set of public partition

**Instagram**. The original dataset for Instagram is provided by [14]. Since the original dataset did not contain graph information, we obtained users' follow lists, personal introductions, and tags for commercial users through Instagram's public API[3]. Therefore, the node text feature for Instagram is the user's personal introduction, and the edge represents the mutual relationship.

**Reddit**. Reddit is constructed on a public dataset [4] that collected replies and scores from Reddit users. The node text feature of this graph is the user's historical post content (limited to the last three posts per user), and the edge represents mutual replies between two users. We divided users into popular and normal categories based on their average score of history posts, with users whose average score is higher than the median considered popular and others considered normal.

## A.2  Prompts

According to the information of the downstream task and graph, this article has designed simple prompts for each dataset. As shown in Table 5, all prompts are added before the node textual features. It should be noted that because PLMs are sensitive to prompts, different prompts may result in significant performance differences. However, how to find suitable prompts is not the focus of this paper, so no search for prompts is conducted.

Table 5: Detailed prompts on three datasets. All prompts are added before node features.

| Dataset | Node feature | prompts |
|---|---|---|
| Arxiv | {abstract} | {This paper is published on [mask] subjection, its abstract is: } |
| Instagram | {profile} | {This user is a [mask] user on Instagram, his profile is: } |
| Reddit | {content of last 3 posts } | {This user is a [mask] user on Reddit, his last 3 posts is: } |

## A.3  Baselines

**PLM-cls**. It represents using the hidden states of RoBERTa-Large directly corresponding to the [cls] token (without any prompts) as node features.

**PLM-prompt-dense**. It represents using the hidden states of RoBERTa-Large directly corresponding to the mask in the prompt (without passing through the final prediction layer) as node features.

**PLM-prompt-sparse**. It represents using the predicted results of RoBERTa-Large corresponding to the mask in the prompt (filtered in the same way as in G-Prompt) as node features.

**GAE**. Its encoder consists of MLP and the input features are the [cls] representations of each node based on RoBERTa-Large (same as PLM-cls). We implement it based on the code provided by PyG[5]. The training epochs are set to 300. The final node feature is the output of MLPs.

**GAE+prompt**. The framework is similar to GAE, but its input features are the prompt representations, namely, PLM-Prompt-dense.

**GIANT**. In Arxiv, we use the pre-trained model provided by the author[6]. As the authors do not provide pre-trained models for Instagram and Reddit, we retrained GIANT on these two graphs using their provided code.

**GIANT+prompt**. We do not modify the training pipeline of GIANT. During inference, all nodes' textual feature is augmented with the same prompts as in G-Prompt and then fed into the GIANT to obtain text features that include the prompts.

---

[3] https://developers.facebook.com/docs/graph-api
[4] https://convokit.cornell.edu/documentation/subreddit.html
[5] https://pytorch-geometric.readthedocs.io
[6] https://github.com/amzn/pecos/tree/mainline/examples/giant-xrt

## A.4 The pipeline of G-Prompt

---
**Algorithm 1:** Training pipeline of G-Prompt

---
**Input:** Node textual feature $\mathbb{S} = \{S_i, i \in V\}$, graph $G = \{V, A\}$
**Output:** The trained parameters of the graph adapter $\Theta^*$
   // Sample training token
**1 for** $i : V$ **do**
**2**      $\hat{S}_i, C_i, Y_i = random\_mask(S_i, mask\_ratio)$// $C_i$ is the position set of masked tokens
**3**      $\hat{H}_i = \mathbf{PLM}(\hat{S}_i)$ // see Eq.(1)
   // Training
**4 for** $epoch : range(max\_epoch)$ **do**
**5**     **for** $i : V$ **do**
**6**         **for** $k : C_i$ **do**
**7**             **for** $j \in Sample(\mathcal{N}_i)$ **do**
**8**                  $\tilde{h}_{i,k,j} = f_\Theta(\hat{h}_{i,k}, z_j)$ // see Eq.(6)
**9**                  $\tilde{y}_{i,k,j} = f_{\mathrm{LM}}(\tilde{h}_{i,k,j})$
**10**                  $\mathcal{L}_{i,k,j} = CE(\tilde{y}_{i,k,j}, y_{i,k})$// see Eq.(8)
**11**                  $backward(\mathcal{L}_{i,k,j}, \Theta)$

**12** $\Theta^* = \Theta$ ;
**13 return** $\Theta^*$

---

---
**Algorithm 2:** inferring pipeline of G-Prompt

---
**Input:** $\mathbb{S} = \{S_i, i \in V\}$, $G = \{V, A\}$, Prompts $P = \{p_1, p_2, ...\}$, $\Theta^*$
**Output:** The node feature $\{x_{i|p}, i \in V\}$
   // Add prompts
**1 for** $i : V$ **do**
**2**      $\tilde{S}_i = Concat(P, S_i)$ ;
**3**      $\hat{h}_{i|p} = \mathbf{PLM}(\tilde{S}_i)$ // see Eq.(3)
   // inferring
**4 for** $i : V$ **do**
**5**     **for** $j \in \mathcal{N}_i$ **do**
**6**          $\tilde{h}_{i|p,j} = f_{\Theta^*}(\hat{h}_{i|p}, z_j)$ // see Eq.(6)
**7**          $\tilde{y}_{i|p,j} = f_{\mathrm{LM}}(\tilde{h}_{i|p,j})$
**8**      $\tilde{y}_{i|p} = \mathbf{MeanPool}(\tilde{y}_{i|p,j} | j \in \mathcal{N}_i)$ ;
**9**      $x_{i|p} = \mathbf{Filter}(\tilde{y}_{i|p})$;
**10 return** $\{x_{i|p}, i \in V\}$;

---

## A.5 Implementation details

**Randomly mask sentences**. For each node in all datasets, we randomly replace 20% of the tokens (textual features) with masked tokens during the training of the graph adapter.

**Framework of the graph adapter**. In E.q (6), the hidden size of $a_{ij}$ is 256, and the MLP layer is set to 2. In the Arxiv, Instagram, and Reddit datasets, the hidden sizes of the MLPs are 7680, 3840, and 3840, respectively.

**Training of the graph adapter**. During each epoch, every saved token will randomly select four neighbors for training. The batch size for the training stage is set to 10,000 pairs. The learning rate is set to 1e-6 and the weight decay is 0.01.

All experiments are conducted on the PowerEdge T640, consisting of 46 Intel Xeon CPUs with 503GB of RAM and 2 Nvidia P100 GPUs with 16GB of memory each.