# OpenReview forum: "Prompt-based Node Feature Extractor for Few-shot Learning on Text-Attributed Graph"
_NeurIPS.cc/2023/Conference — Submitted to NeurIPS 2023_

### Official Review · Reviewer_qmSV · 2023-06-12

**Soundness:** 2 fair
**Presentation:** 2 fair
**Contribution:** 3 good
**Rating:** 4
**Confidence:** 4

**Summary:**

This paper tackles the problem of representation learning on text-attributed graphs (TAGs), which has gained significant attention in recent years. The primary focus of this research is on few-shot node classification. Existing two-stage methods have been unsuccessful in effectively capturing the complex relationship between graph structure and textual features, as well as incorporating downstream task information. To address these limitations, the authors propose a novel approach called G-Prompt, which aims to simultaneously capture the graph topology and downstream task information.

G-Prompt achieves this by enhancing the pretrained language model (PLM) with graph awareness. At the last layer of the PLM, a graph adapter is introduced, allowing the model to become aware of the graph structure. This graph adapter can be combined with task-specific prompts, enabling the generation of task-specific and graph-aware node representations.

**Strengths:**

- Clear motivation of the method
- Well-written summary of existing work (taxonomy, pros and cons)
- Good results in few-shot node classification
- Valuable insights into the utility of graph information and prompts for both LMs and GNN based methods


**Weaknesses:**

- **Some tables and figures are not self-contained, such as Figure 1 and Table 2.** The caption is too concise for the readers to follow. For example, in figure 1, the authors are expected to summarize the proposed framework in one or two sentences to aid comprehension; For table 2, the meaning of the bold and underline should be explicitly specified to ensure clarity and understanding.
- **Background, method, and implementation of few-shot learning is missing.** As few-shot learning is less common in the graph community compared to supervised learning, it is essential to provide additional explanations and context for unfamiliar readers.
Moreover, when comparing to some existing approaches, such as GIANT, which were originally designed for supervised learning, it should explicitly discuss how these methods can be adapted or extended for few-shot learning scenarios. Also, some compared methods, such as GIANT, were intended for supervised learning. How to adopt them for the few-shot learning is not straight-forward and should be introduced.
- **The efficiency of the proposed method is stated but not experimentally proven.**
In section 3.3, the authors claim that training the graph adapters can be achieved with very few computing resources. However, to support this claim, it is necessary to provide a comparison with the compared methods (w.r.t. training parameters, memory usage, training speed, and training time, etc). This empirical evidence would provide stronger support for the efficiency of the proposed method. Furthermore, it is worth noting that the training epoch for the graph adapter is 50 for ogbn-arxiv, while typically fine-tuning it for 3-5 epochs is sufficient[1]. Therefore, it raises concerns whether the additional training epoch may negate the benefits introduced by training only the graph adapter.



**Reference:**
[1] Learning on Large-scale Text-attributed Graphs via Variational Inference, ICLR 2023.

**Questions:**

- According to section 3.4, the node features are fed into subsequent GNNs but in Figure 1, the GNN encoder is not part of the framework, which is a little bit confusing.
- The dimention of the node features need to reduced for practical usage (section 3.4). And it is heuristic to select the optimal M. I am wondering how sensitive the graph prompter is to the value of M, and if there is any discussion on the selection of the optimal M.


**Limitations:**

- The proposed framework is not directly compatible with large language models (LLMs). Nowadays, when people refer to prompt-based methods, they mainly focus on utilizing large language models such as GPT. These models are known for their powerful text modeling and reasoning abilities, which have the potential for performing few-shot learning on text-attributed graphs (TAGs) [2]. However, the framework proposed in this paper is not directly compatible with LLMs, as G-Prompt requires access to LLMs' latent embeddings or logits, which are not provided by models like GPT-3.5 and GPT-4.


**Reference:**

[2] Graph-ToolFormer: To Empower LLMs with Graph Reasoning Ability via Prompt Augmented by ChatGPT. arXiv preprint arXiv:2304.11116 (2023).

---

> ### Author Rebuttal · Authors · 2023-08-10
>
> Thank you very much for reading our paper and affirming the motivation, summary of related work, results, and insights of our paper! Also, thank you for your valuable suggestions and questions regarding our paper! Based on your comments, we have done the following work:
>
> **1. Paper revision:**
> * Figures & Tables: We have revised Figure 1, Table 2 (Weakness 1 & Question).
> * Few-shot setting: We have revised the background introduction on graph few-shot learning, and justification for comparing baselines (Weakness 2).
>
> **2. Add time complexity analysis and experimental report (Weakness 3):**
> * We analyzed time and space complexity and reported runtimes of different baselines.
>
> **3. Add analysis on feature dimensions (Question 2):**
> * We added analysis experiments on varying M columns.
>
> **4. Add experiments on GPT2 (Limitation 1):**
> * We added related experiments based on GPT2 and prospects of combining with LLMs in future work.
>
>
> Here are detailed responses to your suggestions:
>
> **1. Thank you for the suggestions on our figures and tables (Weakness 1 & Question 1).**
>
> We have added more descriptions for the figures and tables. Specifically, in Figure 1, we have included an illustration showing the input to the GraphAdapter, and detailed descriptions of the input and output during GraphAdapter training and inference in the legend. In addition, we thoroughly checked all tables in the paper and added more detailed captions.
>
>
> Here are the detailed responses to each comment/question in English:
>
> **2. Thank you for the suggestions on our few-shot learning setup (Weakness 2).**
>
> * First, let me explain the motivation for our few-shot learning setup: We proposed graph few-shot learning because we found that current TAGs modeling methods like GLEM and GIANT are mostly validated on Ogbn-arxiv and Ogbn-product, while in practice the number of labels may be much smaller. In the real world, the amount of labels is often limited. This motivates us to think about how to do TAGs modeling when the number of labels is small. To formalize this problem, we borrowed the few-shot learning setup from prior works [1], i.e. exploring how to do few-shot learning on graphs. Under this setup, we analyzed the limitations of current methods and proposed G-Prompt. Experiments validate that our method can effectively combine PLMs, and prompts, and integrate graph and task-relevant information to perform few-shot and zero-shot inference on graphs.
>
> * Secondly, we made the following modifications to the paper based on your confusion:
>
> 	- Introduction: Explain that we adopt the few-shot setup in prior works to formalize our problem - how to do TAGs inference when labels are limited.
>
> 	- Related work: Introduce current graph few-shot learning works. Explain that they do not explore combining with TAGs. They focus more on new class addition, and how to do classification. The setup has large differences from ours, so these works cannot be directly compared as baselines.
>
> 	- Experiments: we added GLEM as a baseline. When introducing GIANT and GLEM, we emphasize that these methods lack specific designs catering to the few-shot setup. GLEM and GIANT also have large differences in setup from G-Prompt - GLEM aims to achieve end-to-end training on TAGs, while GIANT and G-Prompt employ self-supervised training, aiming to obtain representations that generalize to various downstream tasks. In the analysis of our results, we emphasize that our method effectively leverages both task-relevant information and graph information for inference through the integration of language model prompting. Notably, our method outperforms both GIANT and GLEM under scenarios with limited labels.
>
> **3. Thank you for the suggestions on the efficiency of our method (Weakness 3).**
>
> Since current TAGs methods adopt different language models, GLEM uses DeBERTa while GIANT uses bert-base-uncased, and GraphAdapter mainly adopts RoBERTa, ALBERT, and GPT2 which are more compatible with prompting. These PLMs inherently have differences in inference speed, so a direct comparison of runtimes is not fair. To enable fair comparison, we analyzed the time and space complexities of the three methods from pretraining and downstream training perspectives. We also added runtime comparisons in the appendix, but these results are just for reference. See more details in our response for Fbx8 and Table 5.
>
> **4. Thank you for the question on the dimension of node features in our method (Question 2).**
>
> Following your suggestion, we conducted analysis experiments on varying dimensions. The results are in Table X. Overall, performance stabilizes after 512. As the result shots, we recommend searching the dimension as a hyperparameter in the practice of G-Prompt.
>
> #### Table 7. Performance of G-Prompt with different dimensions of node features
> |Dimension|256|512|768|1024|1280|
> |:-:|:-:|:-:|:-:|:-:|:-:|
> |Arxiv|0.5854|0.5964|**0.5971**|0.5719|0.5955|
> |Instagram|0.5730|**0.5917**|0.5745|0.5803|0.5811|
> |Reddit|0.6080|0.6141|0.6183|0.6177|**0.6215**|
>
> **5. Thank you for the concern about combining with LLMs (Limitation 1):**
>
> Firstly, we added experiments based on GPT2 and found G-Prompt also adapts it well. Although currently, GPT-3.5/4 logits are not available, there are many open-sourced LLMs like LLAMA2 [2], ChatGLM2 [3], which adopt the same loss as GPT2. So theoretically our method can combine with large models. In fact, we did some experiments on LLAMA2-13b, and the conclusions are similar to GPT2 experiments, but due to the space limit of the paper, we plan to write a separate report on G-Prompt experiments on large models.
>
>
> **We hope our responses have addressed your questions. We look forward to further suggestions and feedback from you.**
>
> **Reference**
>
> [1] Meta-GNN: On Few-shot Node Classification in Graph Meta-learning.
>
> [2] Llama 2: Open foundation and fine-tuned chat models
>
> [3] GLM: General Language Model Pretraining with Autoregressive Blank Infilling

---

> > ### Comment · Reviewer_qmSV · 2023-08-17
> >
> > Thank you for the rebuttal. I will maintain my original scores, as I still hold reservations regarding the final readability and quality of the paper.

---

### Official Review · Reviewer_Fbx8 · 2023-06-22

**Soundness:** 3 good
**Presentation:** 3 good
**Contribution:** 3 good
**Rating:** 4
**Confidence:** 3

**Summary:**

In this paper the authors present a new framework that combines the benefit of Graph models with the Large Language Models. The authors argue that with the current modeling paradigm, the LLMs are trained in a downstream task agnostic fashion although using prompts they could be fine-tuned for specific tasks. Graph networks provide a principled approach of inferencing over structured data (such as over Text-attribute graphs), however there is currently no good mechanism to combine graph inference with LLMs and tune the output from LLMs based on underlying structured data. To alleviate this problem, authors propose G-prompt - a new technique that allows them to incorporate graph structure into LLM fine-tuning. To do so, they fine-tune the last layer of LLM by utilising the graph structure information using the same pooling approach as used in a graph model. They show that their mechanism performs better than vanilla LLM training. They further show that their method is better than SOTA methods for few-shot node classification.



**Strengths:**

1. In this paper authors present a novel technique to combine LLM with graph models.
2. The proposed technique has wider application potential as a number of real world applications rely on structured data.
3. The method beats the current SOTA techniques for few-shot node classification.
4. They achieve comparable to supervised techniques for zero-shot learning, which is impressive.

**Weaknesses:**

1. The proposed technique certainly combined the best of sequence modeling using LLM and structured data modeling using graph networks. It would be interesting to try out a simple technique of flattening the graph data into a sequence and training an LLM with it.
2. It would be good to discuss the computation and efficiency tradeoffs with different experiments.

**Questions:**

1. Have you tried an experiment where you flatten the graph (for instance by appending tokens with some positional information) and fed it into an LLM. How does it compare with your proposed G-prompt work?

**Limitations:**

1. It would be interesting to see how this method performs when applied to large visual models (LVMs). Do we see similar trends?
2. Other than classification, could this technique be used for other semantic understanding tasks such as similarity matching?

---

> ### Author Rebuttal · Authors · 2023-08-09
>
> Thank you for your positive acknowledgment of the innovation in our method, its practical applications, and the experimental results concerning few-shot and zero-shot scenarios. We also greatly appreciate your suggestions and the questions you raised. Based on your feedback, we have made the following improvements:
>
> **1. Introduce a new baseline (Weakness 1 & Question 1):**
>
> We have introduced a new baseline, the "Edge-flattening" and provided details about its implementation, results, and limitations.
>
> **2. Add a discussion of efficiency (Weakness 2):**
>
> We have included additional experimental results that discuss computation and efficiency trade-offs. These results demonstrate that G-Prompt not only outperforms existing methods but also maintains significant advantages in terms of time and space complexity.
>
> **3. Add a discussion of Large Vision Models (LVM) (Limitation 1):**
>
> We have explored relevant literature on LVM (Large Vision Models) and discuss the potential integration of G-Prompt with LVM in future work.
>
> **4. Add semantic matching tasks (Limitation 2):**
>
> Design two few-shot semantic matching tasks on Arxiv and Instagram.
>
> Here are detailed responses:
>
> **1. We appreciate your suggestion of exploring flattening graphs and feeding them into PLMs(Pre-trained Language Models).**
>
> This approach was indeed one of our initial considerations. However, it has certain limitations: (a) Length: Due to constraints on language model inputs, it's challenging to directly input extensive graph information. (b) Efficiency: The same node may be involved in multiple neighbors' features, leading to inevitable duplicate computations. As a response to your suggestion and following our paper's approach, we have defined an "edge-flattening" approach. Specifically, for each edge between two nodes with corresponding text features $S_i$ and $S_j$, we construct a prompt. For example, for Arxiv, the prompt is "[Task prompt], Its' abstract is $S_{i}$. One of its' cited papers' abstract is $S_{j}$." We define the prediction result as $ p_{ij} $. We aggregate all predictions for node i to obtain the node representation $p_i = Pool(p_{ij} | i\in \mathcal{N}(i))$. This approach avoids exceeding the token limit of PLMs compared to using all neighbor features directly. Despite this, exceeding the token limit still occur. In such cases, we truncate $S_i$ and $S_j$ similarly. To provide a better comparison, we introduce the "Prompt-sparse*" baseline, which same as "edge-flattening" but masks $S_j$. Experimental results in Table 4 indicate that "Prompt-sparse-Flatten" performs worse on Instagram and Reddit due to reduced node features, but for instances where "edge-flattening" mitigates missing node information, its performance still surpasses prompts on a single node. This suggests that PLMs possess the ability to understand edge meaning. Since "edge-flattening" have a high time complexity, we conduct neighbor sampling for Arxiv with 5 neighbors. It may affect its performance.
>
> **2. We appreciate your suggestion to discuss time and computation efficiency.**
>
> Comparing time complexities of different methods is challenging due to varying compatible base language models. Therefore, we estimate time complexities as follows: Inference time/space complexity for a single node for the language model is $T_{infer}$ and $J_{infer}$, and training time/space complexity is $T_{train}$ and $J_{train}$. Complexity for non-linear transformations of PLM representations is $T_{MLP}$ and $J_{MLP}$. GIANT's [1] complexity is equivalent to the time required for fine-tuning the PLM, i.e., $O(N \times T_{train})$ for training and $O(N \times T_{infer})$ for inference. GLEM[2] has similar complexity to GIANT. Our approach involves a single inference pass of PLMs, after which all operations are independent of PLMs. GraphAdapter predicts tokens based on a few randomly selected edges for each word, resulting in $O(|S_{all}| \times T_{MLP})$ complexity. $|S_{all}|$ is the total number of training tokens in the TAGs. Hence, our total complexity is $O(N \times T_{infer} + |S_{all}| \times T_{MLP})$. Our primary advantage is independence from $T_{train}$, and $T_{infer}$ can be accelerated by many methods. Considering larger language models where $T_{train}$ >> $T_{MLP}$, our approach holds a significant advantage. In terms of space complexity, our approach doesn't demand loading language model parameters during training, resulting in $O(batchsize \times J_{MLP})$ for G-Prompt compared to $O(batchsize \times J_{PLM})$ for methods involving fine-tuning. Generally, $J_{PLM}$ >> $J_{MLP}$, allowing G-Prompt to accommodate larger batch sizes in memory-restricted GPU environments. We also report efficiency comparisons for reproducibility purposes in Table 5.
>
> **3. We appreciate your question regarding limitations and the integration of Large Vision Models (LVM).**
>
> Due to the time constraints during the rebuttal, we could only discuss the potential LVM integration with G-Prompt. We have carefully investigated the LVMs-related paper, GraphAdapter can combine with many multimodal LVMs. And we intend to include the discussion/citation of LVMs in the papers' future work. Since the characters' limitation of response, we don't provide an extensive explanation in this section.
>
> **4. Thank you for your suggestion regarding the evaluation tasks.**
>
> This suggestion aligns well with G-Prompt's design and the paper's theme. We will design two tasks on Arxiv and Instagram: (a) Paper Title Matching and (b) User Post Matching. We promise we will add this experiment before notification.
>
> We hope our responses have addressed your questions. We look forward to further suggestions and feedback from you. **We eagerly anticipate engaging in further discussions with you regarding the integration of LVMs and the details of the new tasks.**

---

> > ### Author Response · Authors · 2023-08-15
> > **Question about updating your score**
> >
> > I noticed your score dropped from 6 to 4, is there some new concern? I'm very eager to know the reason in order to further improve G-Prompt, and I also look forward to receiving your feedback on the new content I've added in my rebuttal.

---

### Official Review · Reviewer_xGC6 · 2023-07-03

**Soundness:** 2 fair
**Presentation:** 2 fair
**Contribution:** 2 fair
**Rating:** 3
**Confidence:** 4

**Summary:**

The paper feeds manually designed prompts to LLMs to get task-specific text features, instead of BERT-based fixed features. Then a GNN is applied on top of node features for node classification. Few-shot node classification on 3 datasets are conducted for evaluation.

**Strengths:**

1.The motivation is clear and reasonable.

2.Results on few-shot node classification outperform previous methods.

**Weaknesses:**

1.The prompts need to be manually designed, which significantly harms the novelty of this work.  It would be more interesting if the prompts can be automatically and jointly learned with the graph part.

2.The writing need to get improved. For example, Figure 1 failed to help me understand the model design of this work. BTW, the word “random” in this figure is misspelled.

3.Experiments are conducted on a single task (few-shot node classification).

4.Related work is not complete. There have been some work focusing on graph+prompt [1,2].

[1]Gppt: Graph pre-training and prompt tuning to generalize graph neural networks

[2]Graphprompt: Unifying pre-training and downstream tasks for graph neural networks

**Questions:**

Refer to the weaknesses

**Limitations:**

No. Can this work be adapted to link prediction or graph-level tasks?

---

> ### Author Rebuttal · Authors · 2023-08-09
>
> First of all, thank you very much for reading our paper and affirming the problem setting we proposed and the effectiveness of our method. Based on your comments:
>
> * we added discussions about soft prompts when introducing model prompts, including how our method can combine with soft prompts, and why we adopt hard prompts in this paper.
> * At the same time, we thoroughly checked errors in the full text, and designed two new tasks - link prediction and few-shot text matching, to further verify the applicability of GraphAdapter.
> * In addition, we added references to related work on graph + prompts.
>
> Before responding, we would like to briefly recap our paper:
>
> **Motivation:** We find most existing TAGs methods are designed and evaluated on Ogbn-Arxiv and Ogbn-Instagram. However, label amounts are often limited in reality. The applicability of current methods to such scenarios is still unexplored. Therefore, this paper aims to investigate a problem overlooked by the graph community - how to do few-shot/zero-shot inference on TAGs.
>
> **Method:** Inspired by prompts in few-shot zero-shot learning, we propose G-Prompt. Its main goal is to incorporate graph information into the language model prompting process, so the process can acquire both task-relevant information through prompts and capture graph information.
>
> **Experiments:** In the few-shot learning setup, we prove G-Prompt can effectively integrate graph and task prompt information, outperforming SOTA TAGs modeling methods given small samples. We also achieve the first zero-shot inference on TAGs - G-Prompt utilizes graph information and infers node properties better than PLMs alone under zero-shot.
>
> **Insights/Contributions:** (a) We are the first to explore few-shot/zero-shot inference on TAGs. (b) Our proposed G-Prompt can effectively combine with language models. (c) We provide a new set of experimental results on doing zero-shot and few-shot learning on graphs. We believe our results can inspire a new direction of combining graphs and pre-trained language models, i.e. through incorporating prompts, we can train captured graph information through language model losses. We also believe our work can draw more attention to zero-shot and few-shot inference on graphs, and provide insights on integrating graphs with currently booming large models.
>
>
> Here are the detailed responses to your suggestions:
>
> **1. Thank you for the suggestions on our prompts. (Weakness 1)**
>
> First, let me explain why we adopted manually designed prompts. Manually designed prompts are the earliest and most universal and simple form of prompts[2]. Combined with human experience, their effectiveness in few-shot and zero-shot scenarios has been validated in many applications. Therefore, as the first work exploring how language models do prompting on TAGs, and how GNNs can be incorporated into this prompting paradigm, **the core of this paper is whether GNNs can combine with prompts, so we adopted hard templates as the verification means.**
>
> Second, we explain why we did not use soft-prompts or many language model adapters. Indeed, manually designed prompts require human design and have limitations. However, many papers show that **soft-prompts do not perform well under low sample regimes** [1] in NLP. Also, soft-prompts have high time and memory costs in the era of large models. Therefore, considering effectiveness and efficiency, as well as our focus on few-shot learning, we discussed the soft-prompt version in the paper.
>
> Notably, **our method does have the capability to combine with soft-prompts.** A simplest form is directly end-to-end training a soft-prompt on the language model, and then applying G-Prompt based on this soft-prompt. However, how to prompt is not the focus of this work, so we will include this part in our future work.
>
> **2. Thank you for the suggestions on our writing. (Weakness 2)**
>
>  We have redrawn the model figures and thoroughly checked the full text.
>
> **3. Thank you for the comments on experiment types. (Weakness 3)**
>
> Indeed, currently limited by the diversity of TAGs, our method like most existing ones only tested on node classification. However, the two datasets we collected, Reddit and Instagram, do support meaningful link prediction tasks. But constructing suitable datasets and tasks for link prediction requires time, so we can only promise to report link prediction results and experiments before the final notification.
>
> **4. Thank you for the suggestions on our related work. (Weakness 4)**
>
> We added graph+prompt works in the related work. However, it should be emphasized that although these works have overlapping titles with ours, e.g. involving graph and prompt, the actual problems explored differ a lot. Graphprompt works to study how to efficiently fine-tune pre-trained GNNs, while our prompt uses PLMs' prompts, exploring how to incorporate graph information into language model prompting. **To avoid confusion, we highlighted this difference when citing these works.**
>
> **We hope our responses have addressed your questions. We look forward to further suggestions and feedback from you.**
>
> **REFERENCE:**
>
> [1] Fantastically ordered prompts and where to find them: Overcoming few-shot prompt order sensitivity.
>
> [2] Language Models are Unsupervised Multitask Learners.

---

> > ### Comment · Reviewer_xGC6 · 2023-08-18
> >
> > Thanks for the response to my questions. But my main concerns (weakness 1) still hold and thus I'll keep my score.

---

### Official Review · Reviewer_CCnd · 2023-07-05

**Soundness:** 3 good
**Presentation:** 2 fair
**Contribution:** 2 fair
**Rating:** 5
**Confidence:** 2

**Summary:**

The paper presents G-Prompt, a novel framework designed to model Text-attributed Graphs (TAGs) more efficiently. G-Prompt addresses the existing limitations of current methods by combining a graph adapter with task-specific prompts to extract node features, thereby integrating information from both the graph structure and downstream tasks. The graph adapter makes pre-trained language models (PLMs) aware of the graph structure, and the task-specific prompts provide task-related interpretations. Experimental results reveal that G-Prompt outperforms existing methods in few-shot learning and demonstrates robust performance in zero-shot settings. The generated node representations also exhibit high interpretability concerning task performance, indicating the framework's effectiveness in harnessing both graph and task-specific information.


**Strengths:**

1. The paper introduces a framework, G-Prompt, which effectively integrates graph and task-specific information for node feature extraction in text-attributed graphs (TAGs), which addresses a significant gap in existing methods.
2. G-Prompt outperforms state-of-the-art methods on few-shot node classification and performs comparably with fully-supervised baselines in zero-shot settings.


**Weaknesses:**

1. While the G-Prompt framework is innovative, it builds heavily on existing concepts such as PLMs, GNNs, and the use of prompts. The novelty lies more in the combination and application of these techniques rather than in entirely new concepts.
2. The experiments could have been strengthed by utilizing more advanced pre-trained language models, such as ALBERT, DeBERTa, etc., to demonstrate its wide applicability to other PLMs.


**Questions:**

Major: Please address the weaknesses.

Minor: There are several typos and stylistic issues. Please carefully revise your manuscript.

1. Table 1: what is "# Eeges"? might be 'edges'?
2. Figure 1: Romdom mask -> Random mask
3. line 85: fuction -> function
4. line 176: it's -> its


**Limitations:**

The authors have not explicitly discussed the limitations or potential negative societal impacts of their work in the sections of the paper that were processed. While the authors demonstrate the effectiveness of G-Prompt on few-shot and zero-shot learning scenarios, it's unclear how well the framework generalizes to other types of tasks or datasets. The authors could discuss this potential limitation and provide insights to address it.

---

> ### Author Rebuttal · Authors · 2023-08-09
>
> First of all, thank you very much for reviewing our paper and for your positive acknowledgment of the problem setting we presented in our paper, as well as your recognition of the effectiveness of our method. Following your suggestions, we have made the following modifications:
>
> **1. Conduct experiments on ALBERT[1] and GPT2[2] (Weakness 2):**
>
> * We have conducted experiments involving ALBERT and GPT2
>
> **2. Summary out method's innovation (Weakness 1):**
>
> * We have added a summary of the method's innovation in the Method section.
>
> **3. Writing revision (Question 1):**
>
> * We have carefully revised the paper and provided additional descriptions for the relevant experiments.
>
> Here is a detailed response to the suggestions you provided:
>
> **1. We sincerely appreciate your suggestions regarding utilizing more PLMs (Weakness 2).**
>
> We have added related experiments based on ALBERT. DeBERTa was not pre-trained with a mask-token prediction task, so it cannot directly utilize prompts for few-shot zero-shot tasks, which is why it cannot be combined with our method. In addition, we supplemented experiments on GPT2. **The results show our method supports both BERT-like PLMs, and GPT-like PLMs.** This means G-Prompt also has potential to be combined with current state-of-the-art generative large language models (PLMs, Pre-train Language Models). We believe these insights can inspire future work to explore integrating GNNs with open-sourced generative large models (LLAMA2[3], ChatGLM2[4], Qwen[5], etc).
>
>
>
> **2. We are grateful for recognizing the innovation in G-Prompt and for your suggestions on our framework (Weakness 1).**
>
> **Response:** You are right that current G-Prompt involves three components: pretrained language models, prompts, and GNNs. We may have spent too much space introducing GraphAdapter details in the submitted version. So let me re-clarify the core contributions and innovations of G-Prompt: **(a) Problem: Our work is the first to explore how to incorporate graph information into language model prompting**, which is of broad application value (supports few-shot learning and zero-shot inference). **(b) Method: We propose a general algorithm to fuse graph information into language models.** Specifically, we design a GraphAdapter to capture graph information, and then fuse the captured information into PLMs' prompting process through NLP pretraining tasks. To verify the generality of this algorithm and following your suggestion, we added experiments based on ALBERT/GPT2, with consistent conclusions as RoBERTa[6] experiments. G-Prompt supports both BERT-like and generative pretrained models. These results show that the training approach of G-Prompt, also the main innovation of our method, is a generalizable algorithm that can be applied to various language models. **(c) Efficiency: Our method only trains the GraphAdapter, so the whole training only needs one inference pass of the PLMs.** This characteristic has significant advantages over end-to-end training in the current environment where large open-sourced models emerge rapidly. If a stronger language model appears, we only need one inference pass on current TAGss, and can train a new G-Prompt for various downstream tasks using the saved representations. **For example, we supplemented experiments on 2 pretrained language models within the very short rebuttal time using just 2 P100s.**
>
> **Specific Modifications:** We appreciate the reviewer's feedback on this problem. In the new version of the paper, we have added a summary of the innovative aspects of our method at the end of the Methods section, which should provide readers with a clearer understanding of the innovative features and design intentions of G-Prompt.
>
> **3. We sincerely thank you for your suggestions on our writing (Question 1).**
>
> We sincerely apologize that due to our rushed writing, your reading experience was negatively impacted. We have carefully checked the typos and errors you summarized in the questions. We consolidated all issues pointed out by reviewers and thoroughly revised the full text. We modified all parts with unclear expression to make them easier for readers to understand.
>
> **We hope our responses have addressed your questions. We look forward to further suggestions and feedback from you.**
>
> **REFERENCES**:
>
> [1] ALBERT: A Lite BERT for Self-supervised Learning of Language Representations.
>
> [2] Language Models are Unsupervised Multitask Learners.
>
> [3] Llama 2: Open Foundation and Fine-Tuned Chat Models.
>
> [4] GLM: General Language Model Pretraining with Autoregressive Blank Infilling.
>
> [5] Introducing Qwen-7B: Open foundation and human-aligned models.
>
> [6] RoBERTa: A Robustly Optimized BERT Pretraining Approach.

---

> ### Comment · Reviewer_CCnd · 2023-08-21
>
> Thanks for your response. I'll keep my original rating.

---

### Official Review · Reviewer_4Eon · 2023-07-07

**Soundness:** 2 fair
**Presentation:** 2 fair
**Contribution:** 2 fair
**Rating:** 4
**Confidence:** 4

**Summary:**

The paper introduces a new framework called G-Prompt for analyzing text-attributed graphs, which are commonly found in real-world networks. The existing methods for analyzing these graphs have limitations in improving performance when there is limited training data. G-Prompt addresses this issue by combining a graph adapter and task-specific prompts to extract better features from the graph. The experimental results show that G-Prompt outperforms existing methods in classifying nodes with limited training data. It also provides more understandable results and performs comparably to fully-supervised approaches in scenarios where no training data is available.

**Strengths:**

1. The paper addresses an interesting problem setting of learning on text-attributed graphs (TAGs). This problem setting has practical implications and contributes to advancing the understanding of graph-based machine learning.
2. The proposed G-Prompt framework demonstrates good performance compared to baseline methods. It outperforms existing state-of-the-art approaches in node classification tasks, particularly in few-shot learning scenarios.

**Weaknesses:**

1. The proposed method lacks sufficient novelty as it combines existing techniques of graph adapters and prompt-based embedding learning, resulting in limited technical innovation.
2. The ablation study of different modules in the proposed framework is lacking. The authors do not sufficiently analyze the individual contributions and impact of each component. In other words, how do the prompting and the graph adapter raise the performance individually?
3. The description of baselines is unclear as there are no citations or explanations provided. It is unclear what the baseline GAE and GIANT refer to, and this lack of clarity hinders the reader's understanding and evaluation of the proposed method.
4. The paper contains several typographical errors, such as the misspelling of "GIANT" as "GAINT" in line 242 and the misspelling of "random" as "rondom" in Figure 1. These errors suggest a need for thorough proofreading.

**Questions:**

1. The authors are doing classification tasks on all three datasets, however, they apply different evaluation metrics to them (ACC for Arxiv and ROC for Instagram and Reddit). Is there a reason for doing so? It's better to have some discussion here.
2. Could the authors further explain the motivation of the GraphAdapter? How it becomes context-friendly and prompting-friendly?

**Limitations:**

No, didn't see any limitations addressed in the paper.

---

> ### Author Rebuttal · Authors · 2023-08-09
>
> First of all, we sincerely appreciate your careful reading of our paper, your positive affirmation of our research questions, and your recognition of the effectiveness of our proposed methods.
> The summary of our response is as follows:
>
> 1. **GraphAdapter Design  (Question 2):**
>    - We have reorganized and expressed the motivation behind GraphAdapter more clearly.
>    - We have meticulously revised the explanation of "Context-friendly" and "Prompting-friendly" in Section 3.2 to enhance clarity.
> 2. **Suggestions for Method Novelty(Weakness 1):**
>    - In terms of writing, we have emphasized the core contributions of our method in both the introduction and the model section and we have highlighted the novelty of our approach compared to existing methods.
>    - Experimentally, we have incorporated experiments involving ALBERT[1]/GPT-2[2] to demonstrate the generalizability of our framework.
> 3. **Suggestions for Ablation Experiment(Weakness 2):**
>    - In writing, we have rephrased the introduction to the ablation experiments in Section 4.3, providing a more comprehensive analysis of the results. Additionally, we've made it clearer in the main experiment table how it corresponds to the respective ablation experiments.
>    - Experiment-wise, we have included performance results for prompt representations under different prompts, as well as results after G-Prompt processing.
> 4. **Writing Suggestions (Weakness 3 & Weakness 4):**
>    - We have consolidated all the issues pointed out by the reviewer and thoroughly revised the entire manuscript.
>    - We have added references to relevant baselines and explanations to enhance the clarity of our descriptions.
>    - We have made necessary modifications to any unclear expressions to ensure better reader comprehension.
> 5. **Dataset Question (Question 1):**
>    - We have updated the dataset description to include the rationale for using ROC-AUC evaluation on the Reddit and Instagram (ins) datasets.
>
> Here is a detailed response:
>
> **1. Thanks for your question about the motivation behind GraphAdapter.**
>
> **We design GraphAdapter with the following motivations:** (a) to enable few-shot/zero-shot learning using prompts, (b) to allow PLMs(Pre-trained Language Models) to utilize graph information during the prompt process. To achieve this, GraphAdapter requires to utilize graph information and combine it with prompt outputs. We propose using unlabeled text data present in TAGs(Text-Attributed Graphs) to train GraphAdapter using the language model's loss function. However, during training, GraphAdapter only sees existing TAGs' text, while during prompt node representation, the input to GraphAdapter from prompts is unseen. To address this, we need to **(a) preserve the language model's contextual understanding** (so it can interpret the added prompts), and **(b) generalize the learned graph information to unknown words, as prompt content might not appear in existing text data.** Thus, "context-friendly" refers to not disrupting the original contextual modeling of the language model. We achieve this by placing GraphAdapter after the last layer of the transformer. "Prompting-friendly" means avoiding learning token-specific content. For instance, if PLM predicts a masked token is "apple", but its' label is "orange", a direct linear transformation might learn a mapping from "any word" to "orange," leading to overfitting. Therefore, we train on "apple" and its "neighbor influence," determined solely by graph and neighbor features. This way, prompt predictions include both PLM's predictions and neighbor influence.
>
> **2. Thanks for your suggestion regarding the novelty of G-Prompt:**
>
> We apologize that due to our writing, the method's contribution may have been overly focused on network structure. The motivation for our GraphAdapter can be found in the previous answer. Indeed, G-Prompt combines currently popular techniques. But we want to emphasize that **we are the first to propose training GNNs(Graph Neural Networks) as an adapter through pre-training tasks of language models.**  Specifically, our innovations are: (a) How to incorporate graph information into the PLMs' prompting process itself has not been explored in previous works. (b) Our method is a general method that can be combined with various PLMs. To further demonstrate this, we added experiments based on ALBERT and GPT2 during the rebuttal period (See Table 2). The results show that G-Prompt can further improve the few-shot capabilities of prompt representations. (c) G-Prompt can accomplish what previous graph methods have struggled with - zero-shot inference on graphs. We believe these insights about training can inspire more work to try combining PLMs and GNNs to explore few-shot/zero-shot learning on graphs.
>
> **3. Thank you for your valuable feedback on our ablation experiments:**
>
> We conduct additional experiments using multiple prompts to explore the relationship between G-Prompt and prompts. The results are presented in Table 3 while prompt details are in Table 4. These experiments show that task-related information improves prompt representations in downstream tasks. Importantly, G-Prompt's representations correlate positively with original prompt representations and consistently outperform them. This result indicates that G-Prompt is robust for prompts
>
> **4. Thank you for your suggestion of our writing:**
>
> We have thoroughly revised our paper following your suggestion, and we have also carefully revised the whole paper.
>
> **5. We appreciate your question about our dataset.**
>
> We choose ROC-AUC as the main evaluation metric for the Instagram and Reddit datasets due to their imbalanced positive-to-negative sample ratio.
>
> **We hope our responses have addressed your questions. We look forward to further suggestions and feedback from you.**
>
> REFERENCES:
>
> [1] ALBERT: A Lite BERT for Self-supervised Learning of Language Representations.
>
> [2] Language Models are Unsupervised Multitask Learners.

---

> > ### Comment · Reviewer_4Eon · 2023-08-22
> >
> > Thank you for the detailed rebuttal. I'll keep my original score.

---

### Author Rebuttal · Authors · 2023-08-09

To all reviewers,

We sincerely appreciate your affirmation of G-Prompt and your insightful suggestions on its current limitations. Guided by your comments, we have conducted extensive additional experiments to supplement this paper, along with revisions to the content. As there are many changes, we provide an overview of the modifications to the paper:

**1. Report G-Prompt's performance based on ALBERT and GPT2:**

We sincerely appreciate Reviewer CCnd's insightful comments! We conducted experiments to evaluate the effectiveness of G-Prompt based on ALBERT-Large[1] and GPT2-Large[2]. It is noteworthy that there are differences in pretraining tasks between BERT-like models and GPT2. However, the GraphAdapter structure remains unchanged for all models, with only minor modifications to the input and corresponding labels of the loss function. The experiment results demonstrate that: (a) the prompt representation of GPT2 remains effective on TAGs, and (b) GraphAdapter not only works well with BERT-like models but also supports GPT-like models. This further confirms that G-Prompt is a general framework.
#### Table 1. The performance of G-Prompt is based on different PLMs. Each row corresponds to a specific method. Every column lists the performance of the methods in a specific PLM of the dataset. The symbol "-" is used for formatting purposes only. Accuracy is used as an evaluation metric for the task in Arxiv, while AUC is used as an evaluation metric for the other two datasets.

||Arxiv|Instagram|Reddit|
|:-:|:-:|:-:|:-:|
||ALBERT-L\|RoBERTa-L\|GPT2-L|ALBERT-L\|RoBERTa-L\|GPT2-L|ALBERT-L\| RoBERTa-L\|GPT2-L|
|Cls-Embedding|--0.4297-\|--0.5414--\|--N/A--|--0.5407-\|--0.5385--\|--N/A--|--0.5366-\|--0.5236--\|--N/A--|
|Prompt-sparse|--0.5466-\|--0.5784--\|0.5580|--0.5511-\|--0.5721--\| 0.5580|--0.5681-\|--0.5761--\|0.5809|
|G-Prompt|--**0.5589**-\|--**0.5927**--\|**0.5863**|--**0.5680**-\|--**0.5917**--\|**0.5863**|--**0.6010**-\|--**0.6167**--\|**0.5956**|

**2. Add prompt-related ablation experiments:**

(Thank reviewers 4Eon and xGC6 for pointing this out) We tested 5 different prompts on each dataset following the classic prompt exploration work [1]. The prompts fall into three categories: task-relevant (the prompt used in this paper), No task information, and irrelevant. The results are shown in Table 3, with a case corresponding prompts of Arxiv listed in Table 4. It can be seen that (1) prompts containing task information perform better than ones without task information, (2) G-Prompt brings significant gains over different prompt representations, (3) G-Prompt's performance correlates with the prompt representation's performance.

#### Table 2. The performance of G-Prompt is based on different prompts. Each row corresponds to a specific prompt. Every column lists the performance of the prompt in a specific method of the dataset. The details of the different prompts can be found in Table 3. The design of evaluation metrics for different datasets is consistent with Table 1.

|||Arxiv|Instagram|Reddit|
|:-:|:-:|:-:|:-:|:-:|
|Cate|Prompt id|RoBERTa-L\|+G-Prompt|RoBERTa-L\|+G-Prompt|RoBERTa-L\| +G-Prompt|
|Task specific.|0|0.5784\|0.5927|0.5721\|0.5833|0.5761\|0.6167|
|No task information|1|0.5485\|0.5854|0.5522\|0.5686|0.5516\|0.5895|
||2|0.5648\|0.5868|0.5504\|0.5710|0.5665\|0.5861|
||3|0.4944\|0.5794|0.5587\|0.5804|0.5552\|0.5919|
|irrelevant|4|0.5550\|0.5902|0.5444\|0.5686|0.5546\|0.5853|

#### Table 3. Details of the different prompts on different datasets. **[MASK]** represents the masked token. All prompts are added before text features([text]) of the node.

|||Dataset|
|:-:|:-:|:-:|
|Cate|Prompt id|Arxiv|
|Task specific.|0|This is a paper published on the **[MASK]** subject of Arxiv, its abstract is: [text]|
|No task information|1|This is a **[MASK]** paper. [text]|
||2|This is **[MASK]**. [text]|
||3| **[MASK]**. [text]|
|irrelevant |4| My favorite fruit is **[MASK]**. [text]|

**3. Added experiments on flattening graphs.**

Following reviewer Fbx8's suggestions and our paper's approach. We explored whether graph information can be directly incorporated into PLMs for prompting by flattening the graph. We have defined an "edge-flatten" approach. Specifically, this method first concatenates the text of two nodes directly along edges as input into the language model, i.e. predicting based on its own information and one neighbor's. Then predictions are aggregated edge-by-edge from neighbors. Due to the character limit of language models, this method inevitably needs to truncate texts. For a controlled experiment, we added "Prompt-sparse\*", which has the same truncation as "edge-flatten". The results are shown in Table 4. More details can be found in our response to reviewer Fbx8.
#### Table 4. The performance of different methods on three datasets. Each row corresponds to a specific method. Every column lists the performance of a specific method of the dataset. The design of evaluation metrics for different datasets is consistent with Table 1.

|Dataset|Arxiv|Instagram|Reddit|
|:--:|:-:|:-:|:-:|
|Prompt-sparse|0.5784|0.5721|0.5761|
|Prompt-sparse*|*0.5806*|0.5493|0.5577|
|Prompt-Flatten|0.5731 (sample) |*0.5815* |*0.5844*|
|G-Prompt|**0.5927**|**0.5917**|**0.6167**|

#### Table 5. Time efficiency analysis on the Arxiv dataset. Each row corresponds to a specific method. Every column lists the time in a specific method of the dataset.
||SSL stage|Downsteam task|
|:-:|:-:|:-:|
|Methods|s/epoch|min|
|GLEM|N/A| 255|
|GIANT| 1818.52|1.2|
|G-Prompt |1039.01|152|

**REFERENCES**:

[1] ALBERT: A Lite BERT for Self-supervised Learning of Language Representations.

[2] Language Models are Unsupervised Multitask Learners.

[3] RoBERTa: A Robustly Optimized BERT Pretraining Approach.

[4] Learning on Large-scale Text-attributed Graphs via Variational Inference.

[5] Node Feature Extraction by Self-supervised Multi-scale Neighborhood Prediction.

---

> ### Comment · Area_Chair_WtfM · 2023-08-21
>
> Thanks for the rebuttal (here and below for each reviewer). We will take it into account during the discussion period.

---

### Decision · Program_Chairs · 2023-09-21

**Decision:**

Reject

**Comment:**

The reviewers have many concerns about the paper which remained unchanged after acknowledging the author response.